# Boosting the electronic and catalytic properties of 2D semiconductors with supramolecular 2D hydrogen-bonded superlattices

Can Wang[1], Rafael Furlan de Oliveira[1], Kaiyue Jiang[2], Yuda Zhao[1], Nicholas Turetta [1], Chun Ma[1], Bin Han[1], Haiming Zhang [3], Diana Tranca[2], Xiaodong Zhuang [2], Lifeng Chi [3], Artur Ciesielski [1✉] & Paolo Samorì [1✉]

The electronic properties of two-dimensional semiconductors can be strongly modulated by interfacing them with atomically precise self-assembled molecular lattices, yielding hybrid van der Waals heterostructures (vdWHs). While proof-of-concepts exploited molecular assemblies held together by lateral unspecific van der Waals interactions, the use of 2D supramolecular networks relying on specific non-covalent forces is still unexplored. Herein, prototypical hydrogen-bonded 2D networks of cyanuric acid (CA) and melamine (M) are self-assembled onto $MoS_2$ and $WSe_2$ forming hybrid organic/inorganic vdWHs. The charge carrier density of monolayer $MoS_2$ exhibits an exponential increase with the decreasing area occupied by the CA·M unit cell, in a cooperatively amplified process, reaching $2.7 \times 10^{13}$ cm$^{-2}$ and thereby demonstrating strong n-doping. When the 2D CA·M network is used as buffer layer, a stark enhancement in the catalytic activity of monolayer $MoS_2$ for hydrogen evolution reactions is observed, outperforming the platinum (Pt) catalyst via gate modulation.

[1] Université de Strasbourg, CNRS, ISIS, 8 allée Gaspard Monge, 67000 Strasbourg, France. [2] The Meso-Entropy Matter Lab, The State Key Laboratory of Metal Matrix Composites, Shanghai Key Laboratory of Electrical Insulation and Thermal Ageing, School of Chemistry and Chemical Engineering, Frontiers Science Center for Transformative Molecules, Shanghai Jiao Tong University, 200240 Shanghai, P. R. China. [3] Institute of Functional Nano & Soft Materials (FUNSOM), Jiangsu Key Laboratory for Carbon Based Functional Materials & Devices, Soochow University, 215123 Suzhou, P. R. China. ✉email: ciesielski@unistra.fr; samori@unistra.fr

Since the first isolation of graphene[1], the family of two-dimensional materials (2DMs) has expanded rapidly providing access to materials holding different properties. Their combination, by arbitrary stacking dangling-bond-free 2D nanosheets to form van der Waals heterostructures (vdWHs)[2], enabled the further diversification of the properties[3,4] as a result of the charge redistribution at the interfaces of different 2DMs'crystals[5].

Artificial vdWHs can be produced by using diverse strategies, such as the pick-and-lift technique and wet-transfer from chemical vapor deposition (CVD) samples. However, all these methods suffer from the lack of control over the lattice alignment at the interface between two 2D nanosheets. Moreover, the transferred layers can exhibit microcracks, wrinkles, and contaminations by residues from the often-employed sacrificial layers. Epitaxial growth techniques are frequently exploited for the fabrication of 2D heterostructures via lateral overgrowth[6]. Additionally, lattice and thermal expansion coefficient mismatches limit the possibility for growth and integration of high-efficiency electronic and photonic devices based on dissimilar 2DMs. Most importantly, neither of these conventional methods can meet the requirements for programming highly ordered electrostatic superlattices in vdWHs, to ultimately efficiently modulate the electronic properties of 2DMs. In this context, the use of the supramolecular networks represents a simple, yet remarkably effective solution to overcome the aforementioned problems. 2DMs provide atomically flat surfaces for molecular assembly governed by non-covalent interactions[7,8]. The physisorption of suitably designed molecules allows the formation of self-assembled monolayers (SAMs) under thermodynamic control which represents true superlattices on top of the crystalline 2DMs. The presence of electron-donating/withdrawing groups in the molecule employed to form the SAMs can lead to doping of 2DMs through the controlled local modifications of their surface potential[8–10]. Compared to molecules randomly adsorbed on the 2DMs, the use of highly ordered superlattices may maximize the electronics variation via collective or even cooperative effects. Pioneering works demonstrating the feasibility of 2DMs' surface decoration with molecular monolayers include organic/inorganic vdWHs of graphene, h-BN, and MoS$_2$[7]. While the band structures of the pristine layered materials are modified by the additional periodic potential introduced by SAM lattices, hitherto the examples are limited to monolayers formed via unspecific weak intermolecular interactions between alkyl chains that result in small domains (<10$^5$ nm$^2$) featuring random orientations[11–13]. Additionally, SAMs formed via lateral vdW interactions may undergo dynamic rearrangements upon humidity[14,15] or high temperature exposure[16] further leading to instabilities in 2DM-based (opto)electronic devices under operation. The use of 2D supramolecular structures held together by directional and specific non-covalent interactions appears as an ideal solution to attain higher stability and structural control over the micrometer scale.

Intermolecular hydrogen bonds exhibit higher energy (25–40 KJ/mol) compared to intermolecular vdW forces (5 KJ/mol), and they have been widely exploited to guide the self-assembly of suitably designed molecular modules into a variety of supramolecular architectures including 1D, 2D, and 3D arrangements[17]. This type of interaction is unique as it offers a high level of control over the molecular assembly process, along with high specificity, directionality, and reversibility. H-bonding interactions can benefit from a wide range of interaction energies, depending on the number and position of consecutive strong H-bonds. In this framework, melamine (M) represents an archetypal system for the formation of stable non-covalent architectures, in the form of bi-component networks with cyanuric acid (CA)[18], perylene-3,4,9,10-tetracarboxylic di-imide[19], and other di-imide molecules[20,21]. Micrometer-size extended CA·M networks self-assembled on 2D substrates, including HOPG, Au(111), and black phosphorous[22–25], have been shown to exhibit a low amount of structural defects and elevated thermal stability[23].

Here, we study the effect of the physisorbed CA·M bi-component self-assembled network on the electronic properties of the underlying TMDCs. We found the occurrence of a strong charge-transfer process taking place on TMDCs as characterized by a positive cooperative effect emerging in densely packed CA·M bi-component tri-hapto hydrogen-bonded networks. Hybrid van der Waals heterostructures based on molybdenum disulfide (MoS$_2$) functionalized with CA·M supramolecular lattices exhibit an increase in charge carrier density, exceeding $10^{13}$ cm$^{-2}$. Moreover, the same superlattice grown on tungsten diselenide (WSe$_2$) also displays similar significant improvement of the electronic transport through the 2D material. Further insights into the electronic effect of this functionalization approach are gained with density functional theory (DFT) calculations and Kelvin Probe Force Microscopy (KPFM). The efficient charge transfer from the physisorbed CA·M superlattice to the 2DMs determines an upward shift of the Fermi level (E$_F$) of the 2DM. The studies on MoS$_2$ as an earth-abundant and inexpensive material are essential for the implementation of clean energy technologies using hydrogen. The formation of MoS$_2$ based 2D heterostructures, e.g., graphene/MoS$_2$[26] and WS$_2$/MoS$_2$[27], induces a built-in electric field formed among the dissimilar layers, enhancing the MoS$_2$ catalytic performance. The highly crystalline nature of the 2D CA·M network and its significant $n$-doping effect on the 2DMs induces a strong enhancement of the hydrogen evolution reaction (HER) activity of MoS$_2$, surpassing the performance of the standard polycrystalline platinum catalyst. The effect the CA·M superlattice shifts the HER overpotential of MoS$_2$ over 100 mV, yielding a small Tafel slope of 40 mV/dec.

## Results

**Positive cooperative effect on charge transfer**. Cooperativity is a hallmark of molecular self-assembly, with the ensemble behaving differently as a whole from the sum of the isolated individual molecules. DFT calculations revealed that the CA and M co-assembly on MoS$_2$ surface exhibits a positive cooperativity on the charge transfer process.

Tightly packed 2D assemblies of M, CA, and CA·M onto the surface of monolayer MoS$_2$ have been modeled (Supplementary Fig. 1). The different charge density images based on such superlattices (Fig. 1a–c) indicate the charge transfer process from the organic layer to the MoS$_2$, and the calculated magnitude of charge transfer is summarized in Fig. 1d. M molecules are found to adopt a bent conformation in tightly packed monolayers on MoS$_2$, which leads to a lower magnitude of charge transfer ($2.47 \times 10^{13}$ cm$^{-2}$) compared to nearly perfectly flat CA assemblies ($2.70 \times 10^{13}$ cm$^{-2}$). The bi-component CA·M assembly onto MoS$_2$ reveals a charge transfer magnitude as high as $3.62 \times 10^{13}$ cm$^{-2}$, hence outperforming the mono-component assemblies. To gain deeper insight into the role of the admolecules on the local electronics of the MoS$_2$, the magnitude of the charge transfer is computed for mono-component M and CA as well as for the bi-component CA·M superlattice as a function of the tightness of the packing at the supramolecular level (Fig. 1e). We find a linear growth of the charge transfer with the decreasing area of the unit cell of the superlattice for mono-component M and CA assemblies-based hybrids, highlighting a collective nature of the effect[28]. In stark contrast, the bi-component CA·M superlattice exhibits a positive cooperativity

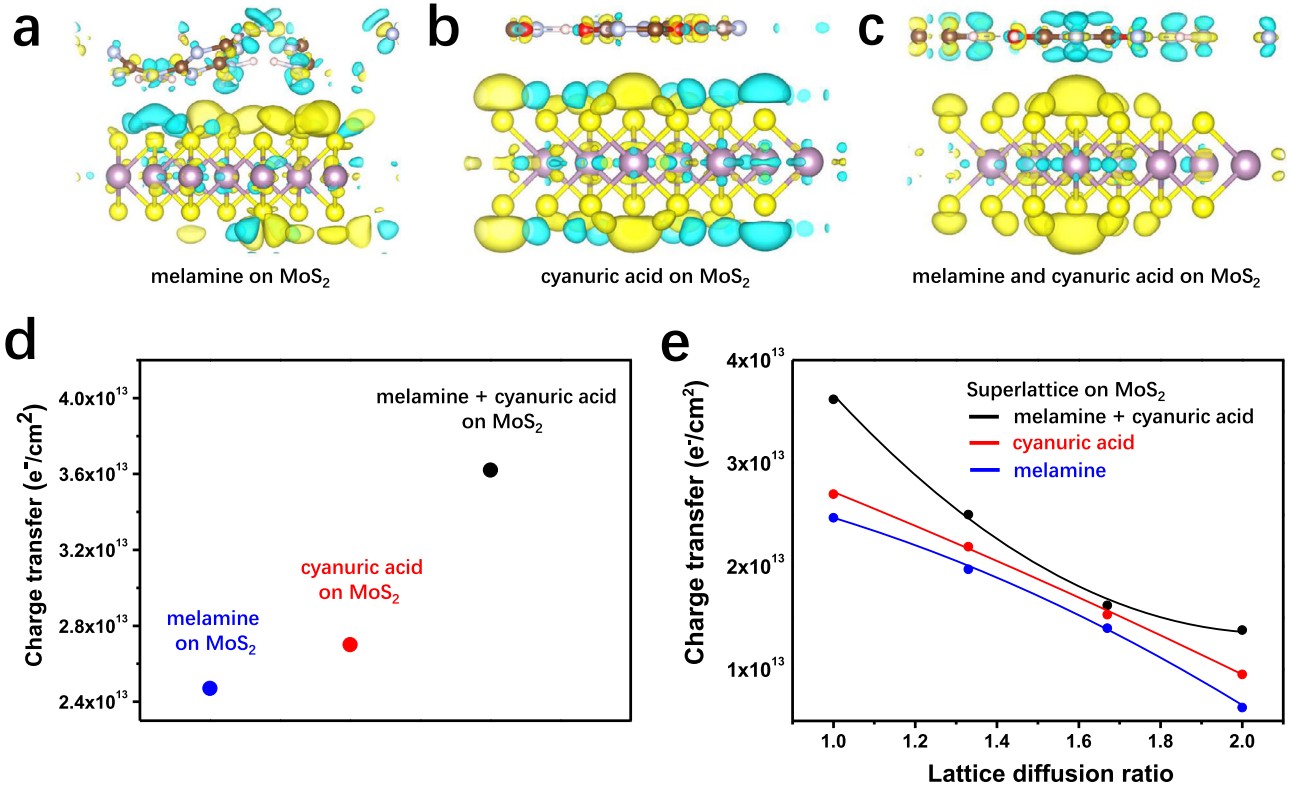

**Fig. 1 Simulated charge transfer from melamine (M), cyanuric acid (CA), and co-assembled structure to MoS₂ monolayer. a–c** The side-view differential charge density images of **a** melamine (M), **b** cyanuric acid (CA), and **c** co-assembled network (CA·M) on monolayer $MoS_2$. The yellow and light blue regions in panels indicate the charge accumulation and depletion, respectively. **d** The magnitude of the charge transfer in tightly packed molecule adlayers. **e** Charge transfer as a function of unit cell size for melamine, cyanuric acid, and co-assembled networks on monolayer $MoS_2$.

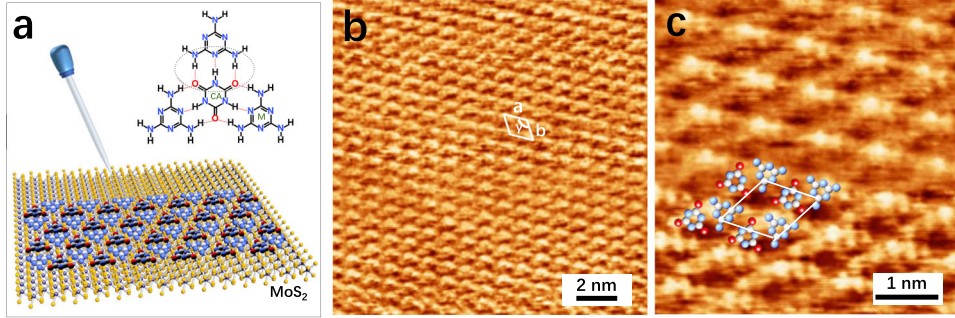

**Fig. 2 Supramolecular network of co-assembled CA·M monolayer onto monolayer MoS₂. a** Schematic representation of the network formation, and the molecular structure of CA·M formed through hydrogen bonding between on monolayer $MoS_2$. **b**, **c** Scanning tunneling microscope height image of co-assembled network adsorbed onto $MoS_2$ surface, and its unit cell is denoted by a white rhombus. Unit cell parameters, i.e., vectors $a = b = 0.96 \pm 0.02$ nm, and the angle between them $\gamma = 120 \pm 2°$ (**b**) survey (**c**) zoom-in with proposed molecular packing motif of CA·M. Tunneling parameters: **b** tip bias voltage $(V_t) = -930$ mV, average tunneling current $(I_t) = 60$ pA; **c** $V_t = -950$ mV, $I_t = 60$ pA.

with an exponential growth of the magnitude of the charge transfer upon reduction of the area of the unit cell. At all values of lattice areas, the magnitudes of the charge transfer for the bi-component CA·M are greater than those determined for the mono-components M and CA superlattices on $MoS_2$.

**Hybrid structures formation and characterization**. The formation of CA·M network on the TMDCs surface is schematically illustrated in Fig. 2a. A 1:1 mixture of CA and M molecules is prepared with Milli-Q water at a concentration of $1 \times 10^{-7}$ M. A 20 µl drop of the warm solution (ca. 350 K) is deposited onto a freshly cleaved $MoS_2$ crystal surface which is subsequently transferred into a desiccator to promote the solvent evaporation.

The modified surface is then rinsed abundantly with water to remove the excess of physisorbed molecules not involved in the formation of the 2D CA·M network, followed by thermal annealing at 430 K in a vacuum for 12 h. Atomic force microscopy (AFM) imaging is employed to monitor the evolution of the surface morphology. It reveals that annealing under vacuum promotes molecular rearrangement on the $MoS_2$ surface yielding a drastic increase in the size of the CA·M domains up to tens of µm² forming a continuous crystalline film (Supplementary Fig. 3b). Such 2D CA·M network exhibits an average thickness of $0.397 \pm 0.103$ nm.

The structure of the dry self-assembled 2D CA·M network is further investigated with a sub-nm resolution by scanning

tunneling microscopy (STM) imaging under ambient conditions. In general, the on-surface molecular assembly is driven by the interplay between molecule-substrate and molecule-molecule interactions. In the present case, the 2D CA·M network formation is mainly governed by intermolecular interaction through the generation of three tri-hapto hydrogen bonds between one CA and three M molecules (Fig. 2a). Such self-assembly pattern further expands across the surface to form a large-area 2D supramolecular structure, where the CA·M network physisorbs onto the underlying MoS$_2$ support as stabilized by vdW interactions. Figure 2b reveals that the CA·M network displays a hexagonal symmetry similar to that of the underlying MoS$_2$ lattice. The unit cell parameters are measured as a = b = 0.96 ± 0.02 nm, and $\gamma = 120 \pm 2°$. As the CA·M structure observed on the MoS$_2$ surface corresponds well to the periodic distance observed in bulk CA·M crystals (measured as 0.964 nm)[29], we consider that the packing structure of the formed 2D CA·M monolayer is completely dominated by the intermolecular interactions. The detailed structural information gathered from high-resolution STM imaging (Fig. 2c) reveals the presence of hexagonal cavities in the CA·M network. DFT simulations of the optimized CA·M structure confirms that the CA·M network adsorbs flat on MoS$_2$ surface, with the CA and M molecules interacting through N-H···O and N-H···N hydrogen bonds (Supplementary Fig. 1c). The simulated SAM structure perfectly matches the STM results.

The same preparation procedure of this hybrid supramolecular adlayer was also applied by using the WSe$_2$ crystal as support, and a similar surface morphology and structure have been monitored by AFM and STM, respectively (Supplementary Fig. 4). Due to the strong multiple hydrogen bonds holding adjacent molecules together, the nature of the underlying surface has little effect on this bi-component assembly. The large-scale highly ordered CA·M monolayer can be easily obtained on the atomically flat surfaces, demonstrating that this should be a universal approach to prepare hybrid TMDC/CA·M vdWHs.

**Electrical and spectroscopic characteristics**. The influence of the 2D CA·M network on the electronic characteristics of TMDCs has been investigated by means of electrical and spectroscopic characterizations. Monolayer MoS$_2$ back-gated FETs (Fig. 3a) are fabricated on n$^{++}$-Si/SiO$_2$ (270 nm oxide thickness) with patterned top Au source and drain electrodes (channel length (L)/ width (W) amounting to 1.77 μm/1.79 μm, respectively). Transfer curves are recorded at source-drain bias V$_{DS}$ = 100 mV before and after the CA·M network deposition to assess its effect on the device electrical characteristics (Fig. 3b). Upon coating with CA·M monolayer, a significant shift of the MoS$_2$ FET threshold voltage ($\Delta V_{th}$) to the negative V$_{GS}$ is observed (Fig. 3b and Supplementary Fig. 5). The off-state current increases from 0 to 1.1 μA at V$_{GS}$ = −80 V. This can be attributed to a strong electrostatic n-doping effect[30]: the functionalized monolayer MoS$_2$ is degenerately doped showing reduced V$_{GS}$ dependence in the transfer curves. The n-doping effect can be quantified from the shift of the $\Delta V_{th}$, which linearly correlates to the charge carrier density change ($\Delta n$). Here, $\Delta V_{th}$ is estimated as 125 V (Supplementary Fig. 5a) after CA·M formation, leading to a $\Delta n$ of $1.1 \times 10^{13}$ cm$^{-2}$ for the transfer curve shown in Fib. 2b. The measurements are reproducible on five different devices, reaching a maximum $\Delta n$ of $2.7 \times 10^{13}$ cm$^{-2}$ and average $\Delta n$ of $(1.9 \pm 0.8) \times 10^{13}$ cm$^{-2}$. The calculated magnitude of the charge transfer matches well with the simulation result, $3.6 \times 10^{13}$ cm$^{-2}$ in Fig. 1a. To our knowledge, previously reported changes in electron density caused by organic molecules physisorbed onto monolayer TMDCs falls usually in the range of $10^{10}$–$10^{12}$ cm$^{-2}$,

even when thick molecular films are used[31–34]. Significantly, our results indicate that when the CA·M molecules are arranged into crystalline H-bonded self-assembled 2D supramolecular structures the doping effect is enhanced. Meanwhile, the field-effect electron mobility ($\mu$) of the CA·M functionalized MoS$_2$ monolayer results ca. 38 cm$^2$ V$^{-1}$ s$^{-1}$, being notably larger than the 14.3 cm$^2$ V$^{-1}$ s$^{-1}$ measured in pristine device. After CA·M coating, the increased electron density induces a decreased contact resistance and screening of Coulomb scatterings, yielding a $\mu$ increase. Further analysis of the device transfer curves indicate that hysteresis decreases from 27.5 V to 8.1 V upon CA·M coating (Supplementary Fig. 5b, c), which is translated into a significant reduction of scattering from the interfacial traps (from $2.2 \times 10^{12}$ cm$^{-2}$ eV$^{-1}$ to $6.4 \times 10^{11}$ cm$^{-2}$ eV$^{-1}$)[35]. This provides the evidence for the screening of coulomb scattering from interfacial traps. Finally, the reversibility of our supramolecular functionalization strategy is confirmed by desorbing the molecules from the MoS$_2$ surface with hot water (>350 K), as evidenced by the FET transfer curves returning to the original state.

Raman and PL spectroscopies are powerful techniques to assess and quantify the electronic properties of 2DMs and monitor fundamental processes like doping[36]. Here, we observed that the Raman signal from out-of-plane vibrations ($A_{1g}$) of monolayer MoS$_2$ is sensitive to the presence of the CA·M superlattices. The $E^1_{2g}$ mode at 386 cm$^{-1}$ and $A_{1g}$ mode at 405 cm$^{-1}$ display a <20 cm$^{-1}$ difference from the bare MoS$_2$ flake, confirming its monolayer nature[37]. CA·M coating results in a 1.3 ± 0.1 cm$^{-1}$ redshift of the $A_{1g}$ peak and nearly no change in the $E^1_{2g}$ peak position (Fig. 3c). Doping in monolayer MoS$_2$ is reported to affect its $A_{1g}$ vibration mode more significantly than $E^1_{2g}$, where $A_{1g}$ redshift is a clear indication of the increase in the surface electron concentration of MoS$_2$[38]. The n-doping of MoS$_2$ upon CA·M coating is also confirmed by PL spectroscopy. For monolayer MoS$_2$, the direct bandgap at K point enables strong PL emission[37,39], as displayed as a black curve in Fig. 3d for the non-coated flake. Two major peaks originate from the spin–orbit splitting of transition metal Mo, viz. the A exciton peak at lower energies (≈1.84 eV) and the B exciton peak at higher energies (≈2.05 eV). Upon CA·M deposition, the MoS$_2$ A exciton peak redshifts of ca. 25 ± 2 meV and undergoes significantly quenching, while the B exciton peak remains unchanged. Our observations agree with the literature for chemical doping of monolayer MoS$_2$ by charge-transfer[37,38]. Similar results from the electrical and spectroscopical characterization of WSe$_2$ flakes confirm that the CA·M network acts as an n-type dopant on monolayer TMDCs (Supplementary Figs. 6 and 7). The reported assembly of H-bonded supramolecular networks as dopant of TMDC monolayers represents a simple, efficient, and reversible strategy to markedly improve device performance for various applications, such as in FETs and HER, as discussed hereafter.

**Doping and surface morphology relationships**. Frequently, the doping of 2DMs is achieved upon physisorption of thick molecular layers[40,41]. Only a few previous works report the use of physisorbed SAMs to adjust the electrical properties of TMDCs, with the best $\Delta n$ values amounting ca. $10^{12}$ cm$^{-2}$ [12,13,32,42]. Here, the use of H-bonded 2D CA·M supramolecular networks yields an improvement of the electron density in monolayer MoS$_2$, with carrier density being at least one order of magnitude superior. To shed light on the role played by the CA·M monolayer on such doping, we studied the morphology of pristine and functionalized device's channel by AFM (Supplementary Fig. 8). The intermolecular multiple H-bonds determine a high stability of the CA·M 2D structure, which grows forming highly ordered molecular film with micrometer-scale crystalline domains, fully

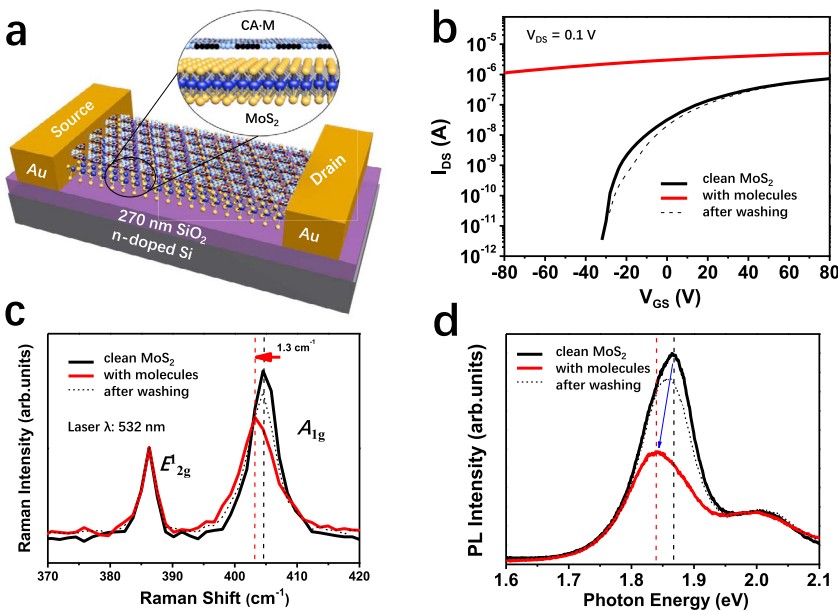

**Fig. 3 Electrical and spectroscopic characteristics of monolayer MoS$_2$ FET with and without the functionalization of melamine and cyanuric acid co-assembled supramolecular network (CA·M). a** Schematics of the MoS$_2$/CA·M hybrid device. **b** Transfer characteristics of MoS$_2$ FET. $I_{DS}$ is the drain current and $V_{GS}$ is the gate potential. **c** Raman and **d** Photoluminescence (PL) spectra of the device before (black curve) and after (red curve) CA·M functionalization, acquired under ambient conditions. As prepared MoS$_2$ samples (black solid), MoS$_2$/CA·M (red solid), device after wiping the CA·M network (black dotted). The peak position is labeled with vertical dashed lines, MoS$_2$ in black and MoS$_2$/CA·M in red. The arrows give the direction of the spectra shift: Raman $A_{1g}$ peak shift in red and PL A exciton peak shift in blue.

covering the TMDC flake's surface. The surface roughness displays a very minor increase upon the physisorption of the CA·M film on both TMDCs, with values of 0.81 nm onto MoS$_2$ monolayer and 0.68 nm onto WSe$_2$ bilayer as quantified on a 1 μm$^2$ area. The defect-free, long-range crystalline structure of CA·M turns out to be crucial to promote such elevated doping. CA·M films produced at lower annealing temperatures (373 K instead of 430 K) onto WSe$_2$ FETs assemble into several smaller crystalline domains (of ca. 0.25 μm$^2$), resulting in a weaker doping effect (Supplementary Fig. 9).

Additionally, we analyze the contributions of the individual components of the supramolecular network, i.e., M and CA molecules on the morphological and electrical characteristics of TMDCs (Supplementary Figs. 10–13). Individual M molecules are observed to self-assemble into multiple small domains on MoS$_2$ (with an average size of $2.5 \times 10^5$ nm$^2$), which are easily disassembled during thermal annealing (Supplementary Fig. 11). On the other hand, individual CA molecules form aggregates on the MoS$_2$ surface at room temperature (Supplementary Fig. 12). From AFM phase images, while the molecular film of individual M covers almost half of the device channel's surface, CA molecules display a greater propensity to undergo aggregation, leaving most of the flake surface uncoated. Such substantial dissimilarity in morphology between mono-component and bi-component films determines a different influence on the electrical characteristics of MoS$_2$ based FETs. For M-only coated devices the negative $V_{th}$ shift is more pronounced than using CA molecules, viz. 28.1 V and 3.4 V, respectively (Supplementary Fig. 13). This leads to $\Delta n$ of $2.2 \times 10^{12}$ cm$^{-2}$ for M-only and $3.7 \times 10^{11}$ cm$^{-2}$ for CA-only coatings on the MoS$_2$ FETs. For measurements taken on five similar devices, the average $\Delta n$ amounts to $(1.9 \pm 0.3) \times 10^{12}$ cm$^{-2}$ and $(4.7 \pm 1.5) \times 10^{11}$ cm$^2$ for M-only and CA-only, respectively, which are both significantly smaller than the values measured on H-bonded CA·M networks with a $\Delta n = (1.9 \pm 0.8) \times 10^{13}$ cm$^{-2}$. The tight packing of CA·M monolayer on TMDC flakes may also

act as an encapsulation layer to protect the device from the environment and subsequent degradation of performance.

**Density of states and Work function modulation.** Further insights into the electronic properties of the MoS$_2$/CA·M hybrid system can be obtained by elucidating the DOS and WF. The doping of a semiconductor shifts its $E_F$ position, ultimately influencing the material's WF[43,44]. Alongside, $E_F$ of a 2D semiconductor is particularly sensitive to modification with organic molecules[45,46]. $E_F$ is known to shift towards the conduction band minimum (CBM) in the case of n-type doping, leading to a lower WF, while p-type doping causes the opposite effect by shifting $E_F$ towards the valence band maximum (VBM).

The evolution of the WF of a monolayer MoS$_2$ surface is measured by Kelvin probe force microscope (KPFM), which allows to record a contact potential map. As shown in Fig. 4a, d, freshly evaporated Au electrodes are used to ground the single MoS$_2$ flakes deposited on a 270 nm SiO$_2$ substrate. Darker regions in the KPFM images (Fig. 4b, e) correspond to larger WF values, indicating that the WF of both pristine and CA·M-doped MoS$_2$ surfaces is smaller compared to the Au electrodes, as shown by the brighter contrast of the flakes. The contact potential difference related to Au increases from 50 mV to 170 mV with CA·M coating. The presence of the CA·M network decreases the WF by 120 meV compared to the pristine MoS$_2$ surface, evidencing a n-doping effect, as determined by averaging the SP values on five monolayer MoS$_2$ flakes and six CA·M coated monolayer MoS$_2$ samples (Supplementary Table 1). Note here that SP differences of nano-objects determined via amplitude modulation KPFM are usually underestimated due to the finite size of the probe, which limits the spatial and potential resolution of the measurement[47]. The calculated WF values (Supplementary Fig. 14a) revealed a negative shift in WF by 220 meV, in good agreement with the KPFM results. These results indicate that due

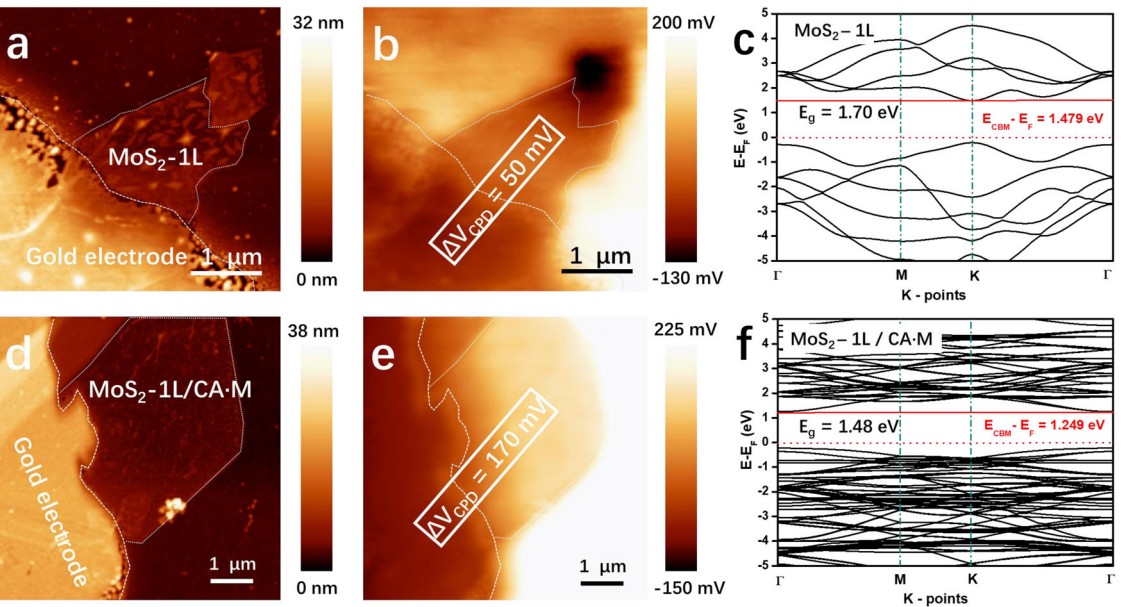

**Fig. 4 Morphology and energetics of coated and uncoated MoS₂ monolayers.** The area of monolayer MoS₂ flake (1L-MoS₂) is circled by white dotted lines. **a** Atomic force microscope (AFM) height image and **b** Kelvin probe force microscope (KPFM) image of a monolayer MoS₂ flake. **c** Calculated electronic band structure of monolayer MoS₂. **d** AFM image and **e** KPFM image of a CA·M doped MoS₂ monolayer flake. **f** The band structure of monolayer MoS₂/CA·M heterostructure. Contact potential difference ($\Delta V_{CPD}$) is the contact potential difference between the $E_F$ of the flake and that of the Au electrode ($E_{F,MoS2} - E_{F,Au}$)/e. The energy level of conduction band minimum ($E_{CBM}$) is marked with red solid lines, and the Fermi level ($E_F$) is marked with the red dashed lines. $E_g$ is the energy gap between conduction band minimum and valence band maximum.

to the interaction between the CA·M and MoS₂ surface, the removal of an electron from the whole hybrid system is much easier than from the CA·M SAM or MoS₂ surface separately.

The DOS simulations are also in good agreement with the literature[44] and corroborate the experimental results, proving that the CA·M network determines n-type doping on monolayer MoS₂ surface. The band structure maps are shown in Fig. 4c, f: the energy gap from $E_F$ to CBM decrease from 1.479 to 1.249 eV after introducing the CA·M SAM. The upward shift of $E_F$ leads to an increased probability of exciting electrons to CBM. The n-type doping on MoS₂ mostly originates from the charge transfer effect of CA·M molecular network, which can be read from the partial density of states (PDOS) plots-derived energy level map (Supplementary Fig. 15).

The MoS₂ monolayer flakes coated with the individual molecular components (either M or CA) were also measured via KPFM, as depicted in Supplementary Fig. 17. Coating with M causes a reduction of the WF of pristine MoS₂ of $\Delta WF = -80$ meV, thus being of smaller magnitude compared to the CA·M system. The WF does not change significantly in CA-coated flakes because of the propensity of the molecules to undergo strong aggregation. When coated with a CA·M superlattice the bilayer WSe₂ displays a negative WF shift of 95 meV (Supplementary Fig. 18), which in good agreement with the simulated monolayer WSe₂, exhibiting a $\Delta WF = -170$ meV (Supplementary Fig. 14b). The experimental WF results from KPFM are summarized in Supplementary Table 1.

**High-efficiency HER catalytic performance.** Electrochemical water splitting is considered one of the most promising approaches towards the production of hydrogen fuel. Platinum (Pt) is so far the best performing and most employed electrocatalysts for the hydrogen evolution reaction (HER)[48], yet its high cost and scarcity greatly limit large-scale applications. Among various possible alternatives to Pt, MoS₂ – material abundant in nature -

exhibits Gibbs free energy adequate for atomic hydrogen adsorption[49]. Although the MoS₂ catalytic activity has been significantly improved over the last decade[50,51], it is still far from the HER performance of Pt. The catalytic activity of MoS₂ is known to be affected by its interaction with supporting substrates via charge transfer[52,53]. CA·M co-crystal nanosheets have been reported as supports for CoP nanoparticles catalyst, giving embedded composite with high hydrogen evolution activity and stability[54]. Since the CA·M network has been proved as a strong n-dopant to MoS₂, it is also expected to act as a suitable buffer layer to boost the catalytic performance of MoS₂. In HER, the adsorbed protons sequester electrons from the MoS₂ surface thereby forming hydrogen molecules through a process of desorption from the surface. Therefore, the electron density on the MoS₂ surface can indirectly affect the rate of hydrogen evolution to a certain extent. The electron transfer in the CA·M/MoS₂ heterostructure occurs from CA·M to MoS₂, hence increasing the charge density on MoS₂ surface and further enhancing the HER activity. Multilayer CA·M film is prepared by drop-casting 200 μL of CA and M mixture solution at $1 \times 10^{-6}$ M concentration on a 1 cm² 270-nm-thick SiO₂ wafer, and then annealed overnight under vacuum at 450 K. The obtained CA·M multilayer film exhibits roughness of 0.29 nm, allowing the easy deposition of mechanically exfoliated monolayer MoS₂ flake on its surface (Supplementary Fig. 19). Raman and PL characterizations confirm that upon inverting the position of the supramolecular network with respect to MoS₂ (i.e., MoS₂ on CA·M) a strong n-doping effect is still induced on the upper MoS₂ monolayer.

The typical three-electrode electrochemical system is employed to evaluate the HER performance of CA·M/MoS₂. A micron-sized window is opened on a photoresist passivation layer on the device surface to expose only the MoS₂ flake to the electrolyte (0.5 M H₂SO₄). Here, monolayer MoS₂ flake is used as working electrode (WE), while Ag/AgCl and pencil graphite are used as reference (RE) and counter (CE) electrodes, respectively (see Fig. 5a and Supplementary Fig. 20 for details). The catalytic performance of

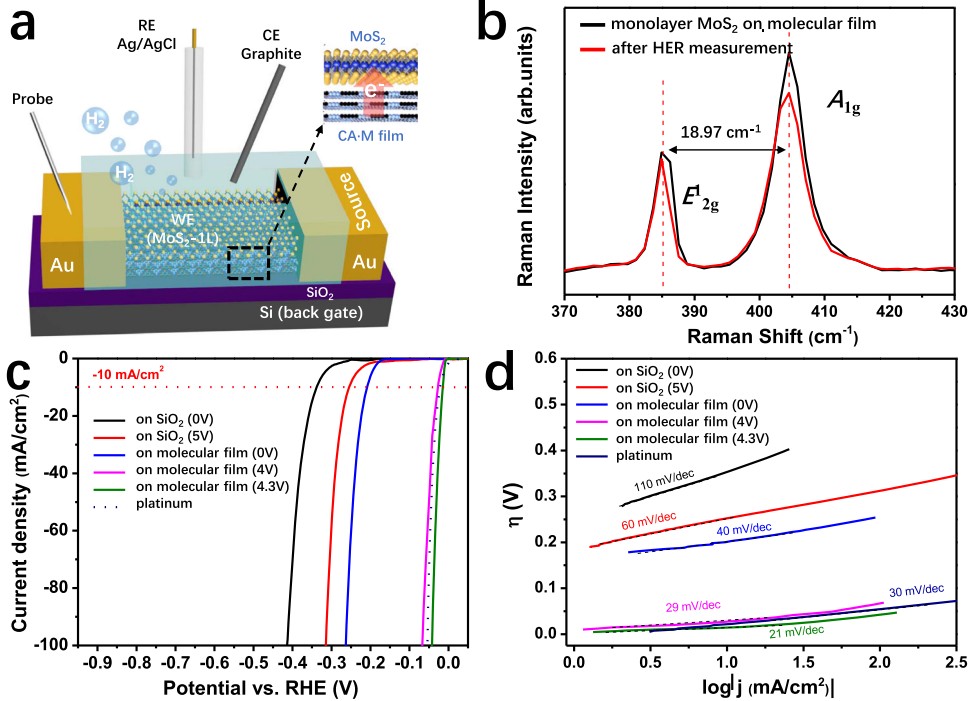

**Fig. 5 Hydrogen evolution reaction (HER) in the co-assembled supermolecule network (CA·M)/MoS₂ heterostructure. a** Monolayer MoS₂ on CA·M molecular film device for HER measurements. The red arrow indicates the charge transfer direction in the heterostructure. **b** Raman spectra of the monolayer MoS₂ flake on CA·M before (black) and after (red) HER measurements, and Raman peak positions are marked by the dashed lines. **c** The polarization curves for HER on Platinum (dashed blue line) and monolayer MoS₂ flakes under different conditions (SiO₂ substrate with/without 2D CA·M buffer layer, and the respective voltages applied gate). The −10 mA cm⁻² current density position is marked by the dashed red line. **d** Corresponding Tafel curves.

neat monolayer MoS₂ flake shows a clear improvement upon addition of the underlying CA·M thin film, as exhibited in Fig. 5c. The HER overpotential ($\eta$) is defined as the potential at the standard current density (j) $-10\,\text{mA}\,\text{cm}^{-2}$. Here, the CA·M/MoS₂ hybrid device reaches $\eta = 208$ mV, ~130 mV smaller than that for the pristine MoS₂ surface ($\eta = 340$ mV), indicating a great enhancement of energy conversion efficiency. The participation of CA·M film improves the electrical coupling between the substrate and the MoS₂ catalyst, therefore the injection of electrons from the electrode and their transport to the catalyst active site is facilitated. Moreover, the CA·M strong *n*-doping effect induces an excessive negative charge density on the MoS₂ surface that boosts the HER performance. Figure 5d depicts the Tafel slope, an important parameter that quantifies the amount of additional applied potential is required to observe a logarithmic increase in current density during HER. The CA·M/MoS₂ hybrid device shows an Tafel slope of 40 mV/dec, which is significantly superior to the values reported for most 2H MoS₂ related compounds (typically 41–120 mV/dec)[55], and even comparable with that for metallic 1 T MoS₂ (43 mV/dec)[56]. The integrity of the MoS₂ flakes is confirmed by Raman spectroscopy carried out after the HER measurements, which endorses the material as a very stable high-performing catalyst.

The conductivity of the MoS₂ channel undergoes a sharp increase when a positive $V_{GS}$ is applied to the FETs, which is considered being a powerful strategy for enhancing the HER catalytic performance[57,58]. The FET characteristics of CA·M/MoS₂ validates that the bottom multilayer CA·M film can induce significant electron density increases on MoS₂ upon back-gate $V_{GS}$ modulation (Supplementary Fig. 21). Moreover, when MoS₂ surface is immersed in a sulfuric acid electrolyte, the applied positive gate voltage has also been confirmed to induce the excess electron density on the Mo atoms near the S vacancy sites, leading

to enhanced Mo-H bond strengths (stabilizing H proton adsorption)[59]. The HER catalytic property of our field-tuned CA·M/MoS₂ devices is largely boosted with $V_{GS}$ increasing (Supplementary Fig. 23c). In particular, the CA·M/MoS₂ channel achieves a better performance than standard polycrystalline Pt catalyst when the CA·M/MoS₂ vdWHs is gated at $V_{GS} = +4.3$ V, exhibiting a $\eta = 14$ mV (20 mV for Pt) and a small Tafel slope of 21 mV/dec (30 mV/dec for Pt). Interestingly, the field effect on the bare monolayer MoS₂ surface is much weaker than on the CA·M/MoS₂ hybrid surface. When applying $V_{GS}$ from 0 V to +4 V, the over potential decreases by 62 mV for monolayer MoS₂ and by impressive 180 mV for the CA·M/MoS₂ hybrid structure (Fig. 5c). The strong *n*-doping effect from the underlying CA·M film enables directional electron flow to MoS₂ active sites, which determines the excellent catalytic performance of this hybrid structure to produce molecular hydrogen. Such a strategy of utilizing the electric field to improve the catalytic activity of HER could also be used in the other electrochemical processes. However, the use of more complicated four electrodes system in cyclic voltammetry experiments may hinder to some extent this gate-enhanced HER strategy for large-scale technological applications.

## Discussion

vdWHs based on 2DMs is attracting ever-growing attention due to their unique electronic features that can be engineered on demand. In this context, the use of the small organic building blocks capable of forming supramolecular lattices through non-covalent interactions represents a simple, yet effective solution towards the modulation of 2DM electronic properties. Here, we show that the bi-component, hydrogen-bonded CA·M 2D network can be used for boosting the electron density of TMDCs through a positive cooperative effect in the charge transfer

process as a function of the packing density of the molecular adlayer. We have found that the physorption of a tightly packed 2D supramolecular network of CA·M onto the monolayer $MoS_2$ channel determines an electron transfer up to $2.72 \times 10^{13}$ cm$^{-2}$, thereby reaching state-of-the-art performance compared to the reported molecule-TMDCs hybrid devices. Optical and electrical analyses corroborated with DFT calculations reveal that the CA·M 2D network induces a charge transfer effect to $MoS_2$, regardless if placed on top or underneath the 2DMs. Owing to the excellent charge transfer effect, CA·M 2D network is used as a substrate to enhance the HER catalyst performance of monolayer $MoS_2$. Remarkably, the field-effect enhanced HER performance of the CA·M/$MoS_2$ hybrid device outperforms the catalytic efficiency of Pt. 2DMs decorated with supramolecular 2D networks may expand the library of heterostructures with on-demand electronic characteristics. Moreover, such an approach can be further employed in multilayer vdWHs displaying electrical, optical, and magnetic functionalities. Compared with classical vdW 2DMs heterostructures, the structural programmability of 2D supramolecular networks enables optimal matching of the crystal structures to form ad hoc superlattices. Importantly, supramolecular chemistry can offer a choice of 2D networks and patterns, with tunable electronic properties, periodic potentials, and atom packing density. The atomic precision achieved through controlled molecular self-assembly can therefore be instrumental for the optimization of charge carriers in 2DMs, and be used to impart novel properties, enabling the realization of multifunctional, high-performance optoelectronic devices.

## Methods

**STM and AFM**. STM measurements were carried out by using a Veeco scanning tunneling microscope (multimode Nanoscope III, Veeco). Bulk $MoS_2$ and $WSe_2$ crystals (HQ graphene) were used as substrates before and after CA·M coating. The employed STM tips were mechanically cut from a Pt/Ir wire (80/10, diameter 0.25 mm). The images were obtained at room temperature in air, and the drift of the piezo was corrected using the underlying chalcogen atomic lattice as a reference. AFM imaging was performed using a Bruker Dimension Icon set-up operating in air, in tapping mode, by using tip model TESPA-V2, tip stiffness K is 42 N m$^{-1}$.

**Raman and photoluminescence spectra**. Raman spectra were carried out in air by Renishaw inVia spectrometer equipped with 532 nm laser, and the wavenumber resolution was 1 meV. The Si Raman peak at 520.5 cm$^{-1}$ was used for normalization. The PL spectra were acquired with a Renishaw InVia spectrometer equipped with a 532 nm laser. TMDC flakes were mechanically exfoliated from crystals onto SiO$_2$/Si substrate. The excitation power was kept below 1 mW to avoid the local heating and damaging effect.

**FET fabrication and characterization**. $MoS_2$ and $WSe_2$ flakes were mechanically exfoliated down to monolayer by the Scotch-tape-based method and further transferred to CA·M/SiO$_2$ and bare SiO$_2$ substrates (270 nm SiO$_2$ thickness onto $n$-doped Si from Fraunhofer IPMS, Germany). Top-gated FET devices were fabricated by maskless optical lithography, with Au source and drain electrodes thermally evaporated onto the patterned substrate. Warm acetone was used for the lift-off process, and the as-fabricated devices were annealed at 430 K in vacuum to remove atmospheric adsorbates. All the FET devices were measured in an N$_2$-filled glovebox with a probe station connected to a Keithley 2636 A source-meter unit.

The carrier mobility μ was determined from the following Eq. (1):

$$\mu = \frac{dI_{DS}}{dV_{DS}} \times \frac{L}{WC_i V_{DS}} \qquad (1)$$

$L$ and $W$ are the channel length and width, and $C_i$ is the insulator capacitance per unit of area.

The dopant-induced charge carrier density increase ($\Delta n$) was calculated from Eq. (2):

$$\Delta n = C_i \frac{\Delta Vth}{e} \qquad (2)$$

Where $\Delta V_{th}$ is the difference in threshold voltage ($V_{th}$) before and after doping effect, and $e$ is the elementary charge ($1.6 \times 10^{-19}$ C).

**KPFM**. Topography and surface potential images were simultaneously collected with Pt/Ir coated silicon probes (Bruker SCM-PIT-V2, resonant frequency ≈75 kHz, $k \approx 3$ N·m$^{-1}$) at ambient conditions using a Bruker Icon AFM employed in amplitude modulation KPFM (AM-KPFM) mode. For work function (WF) referencing and electrical grounding purposes, a polycrystalline Au electrode was thermally evaporated on the flakes, before and after the molecule adsorption. The reported surface potential values are referred to the average SP of the non-treated Au electrode (0.16 ± 0.10 eV) obtained from >30 samples. The calibration of the Au WF (4.85 ± 0.09 eV) was determined on the polycrystalline Au electrodes via macroscopic Kelvin Probe (KP) at ambient conditions (Ambient Kelvin Probe Package from KP Technology Ltd, 2-mm-diameter gold tip amplifier)[60]. The calibration of the KP probe was performed against a freshly cleaved HOPG surface (4.475 eV). The SPM image processing has been done on WSxM software.

**Device fabrication for HER measurements**. Devices for HER measurements were produced by spin coating a layer of photoresist (AZ1505) onto monolayer $MoS_2$ transferred to n$^{++}$-Si/SiO$_2$ substrate. A window was opened on the photoresist by mask-less optical lithography to expose the desired flake area to the solution during measurements. The HER catalytic performance of monolayer $MoS_2$ was measured in a three-electrode configuration in 0.5 M H$_2$SO$_4$ electrolyte (purged with Ar gas), with an Ag/AgCl reference electrode and pencil graphite counter electrode (diameter 3 mm). The $MoS_2$ flake acts as the working electrode during linear sweep voltammetry (LSV) from 0.1 to −0.6 V, with a scan rate of 3 mV s$^{-1}$. All LSV curves were collected after 5 times scan in the same potential range. HER measurements were also performed while biasing the n$^{++}$-Si gate electrode, as depicted in Fig. 4a. The electrochemical current density is calculated by normalizing the current to the area of the exposed window on the $MoS_2$. The data has been referenced to $E_{RHE} = E_{Ag/AgCl} + 0.210$ V.

**DFT calculations**. First-principles calculations within the density functional theory (DFT) framework were performed to investigate different types of self-assembled monolayers (CA and M) adsorbed on the $MoS_2$ surface structures. By means of DFT calculations the adsorption of CA·M SAM on the different surface structures was investigated. Here an important role was played by the small distance (~3.3 Å) between the SAM and the surface, which can influence the electronic properties of the whole system. The WFs were simulated by DFT. In order to identify the most stable configurations, we performed first-principles total-energy calculations using the gradient corrected (PBE) density functional theory as implemented in the Vienna ab initio simulation package[61,62]. The geometry of the $MoS_2$/CA·M hybrid structure was optimized by relaxing the atomic positions of all the atoms. The equilibrium geometry was assumed to be reached when the forces on the relaxed atoms of the system were less than 0.025 eV/Å. A (2 × 2 × 1) Monkhorst–Pack k-point mesh has been used to span the surface Brillouin zone. The system was modeled by using a periodic supercell of 9.81 × 9.81 × 18 Å$^3$. The bond length parameters as well as the unit cell of the structure were chosen to fit the experimental observations. The electronic wave functions were expanded into plane waves up to an energy cutoff of 400 eV, and a projected-augmented-wave scheme[63] was used in order to describe the interactions between the valence electrons and the nuclei(ions).

## Data availability

The data that support the findings of this study are available from the corresponding authors on request.

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

## Acknowledgements

We acknowledge funding from European Commission through the ERC project SUPRA2DMAT (GA-833707), the Graphene Flagship Core 3 project (GA- 881603), and the Marie Curie ITN project UHMob (GA-811284), the Agence Nationale de la Recherche through the Labex projects CSC (ANR-10-LABX-0026 CSC) and NIE (ANR-11-LABX-0058 NIE) within the Investissement d'Avenir program (ANR-10-120 IDEX-0002-02), the International Center for Frontier Research in Chemistry (icFRC) and the Institut Universitaire de France (IUF).

## Author contributions

The experiment was conceived by C.W., L.C., A.C. and P.S.; C.W. performed the AFM, STM characterization; C.W. and Y.Z. carried out FET device study and Raman characterization; K.J., D.T. and X.Z. performed the DFT calculations; C.W., R.F.O. and H.Z. performed the HER measurements; N.T. worked on KPFM; C.M. worked on the capacitance measurement; B.H. performed the Au electrode transfer; P.S. and A.C. supervised the entire study. C.W. wrote the manuscript with comments and suggestions from all co-authors.

## Competing interests

The authors declare no competing interests.
