## [Peer Review File · Nature Communications]

REVIEWER COMMENTS

Reviewer #1 (Remarks to the Author):

This manuscript reported that the monolayer hydrogen-bonded 2D network of cyanuric acid (CA) and melamine (M) can be adsorbed on MoS₂ and WSe₂ to form a hybrid organic/inorganic van der Waals heterostructure. By virtue of the ultra-strong n-doping effect of CA-M, MoS₂ showed an exponential increase in charge carrier density and a higher field-effect electron mobility. Moreover, the hybrid heterostructure can be applied to acidic HER, which greatly improved the activity of MoS₂ (even better than the benchmark Pt). Although the application of CA-M hydrogen-bonded layers to enhancing activity of HER catalysts have been reported (Adv. Mater. 2021, 33, 2007368) but not discussed or cited, this work mainly focused on the fields of semiconductors and FET, which has enough novelty for publication in Nat. Commun.

1. The thickness of CA-M SAMs was claimed as 0.48 nm. However, according to crystallography, the thickness of monolayer CA-M should be about 0.31 nm. Revisions and/or reasonable explanations are needed.
2. It is better to directly characterize the valence state of the materials to support the claim of n-doping. Besides, a deeper explanation for enhancement of HER activity after n-doping should be given.
3. It is better to clearly explain the advantages and disadvantages of applying a gate voltage for HER, particularly in the viewpoint of practical application.

Reviewer #2 (Remarks to the Author):

This paper reports and apparently important result for electrocatalysis, namely that formation of an MoS₂/organic van der Waals heterostructure (VDWH) allows a dramatic reduction in the overpotential for the hydrogen evolution reaction (HER) on the MoS₂. However, I have numerous criticisms of this paper and I do not think it is of suitable quality for Nature Communications. My specific comments are below.

1. The voltammetry shown in Figure 5c is indeed impressive. However, the authors offer no mechanism for enhancement of electrocatalytic activity...they simply ascribe it to excess electron density. That idea alone is not a mechanism or an explanation. The idea that putting electrons into MoS₂ using a gate stack has been published before. Indeed the authors note reference 54 which is the first report of the field effect enhancing HER at MoS₂ sheets. However, the conclusion of reference 54 is that the reason for the enhancement of activity is simply that the contact resistance between the metals and the MoS₂ is reduced on gating...this is a rather mundane result. Likewise reference 56 makes the same conclusion...this time that electrolyte gating of an MoS₂ flake lowers the contact resistance between the metals and the MoS₂. Neither paper argues for an INTRINSIC increase in the activity of the active sites on MoS₂ (S vacancies). The authors should instead take a look at Y. Wang 2019, 19, 6118...that paper proposes an explicit mechanism for HER activity increase with gating (along with experimental results). That paper is directly relevant to the authors' current work. Their results should be compared to it.
2. The presentation of the paper is very frustrating. The introduction does not read well...and even though the HER result is really the most significant finding, there is no hint of this result in the title of the paper.
3. What is the mechanism for the charge transfer between the organic layer and the MoS₂? At the

top of page 10 the authors state that "The n-type doping on MoS₂ mostly originates from the charge transfer effect of the CAM molecular network" but what does that mean? What molecule gets oxidized M or CA? Would M alone dope the MoS₂? Why not?

4. CPD vs SP. In Kelvin probe, the difference in work function between the tip and the sample is called the contact potential difference, not the surface potential. The authors use CPD in Fig. 4 and then the text speaks of surface potential. This should be changed.

5. The point on cooperative effects is not clear enough. Why does formation of the 2D molecular assembly allow doping of the MoS₂ substrate?

Reviewer #3 (Remarks to the Author):

The manuscript by Wang and co-workers reports the formation of inorganic/organic van der Waals heterostructures based on MoS₂ functionalized with cyanuric acid-melamine (CA-M) based extended 2D supramolecular superlattices. It shows that the interfacing of the supramolecular self-assembled network of CA-M with MoS₂ determines a strong n-doping via charge transfer as evidenced by markedly high increase in charge carrier density, exceeding 10^{13} cm⁻². Such a charge transfer from the physisorbed CA-M 2D networks to the MoS₂ and/or WSe₂ determines an upward shift of the Fermi level of the two-dimensional material. Not surprisingly, the CA-M layer has been also found to determine a significant improvement in the electronic transport through the WSe₂. The experimental data has been corroborated by DFT calculations.

Although, the idea of forming 2D self-assembled monolayers based on cyanuric acid and melamine is not new (following the pioneering works of Whitesides in 1990, as detailed by the authors), its use – via controlled interfacing – for modulating the electronic properties of 2D semiconductors is unprecedented. The reported findings open many perspectives on the use of a variety of chemically tailored and atomically precise 2D bi- and multi-component supramolecular superlattices which have been reported during the last 4 decades for tuning the electronic properties of 2D semiconductors via controlled doping.

From a more fundamental viewpoint, is extremely interesting that the while monocomponent networks of melamine and cyanuric acid mildly affect the electronic properties of the 2D semiconductors, the bi-component CA-M network causes major changes in the charge transfer in a process featuring a cooperative nature. This unique phenomenon was clearly unraveled computationally.

I was also impressed by the performance of the CA-M modified 2D materials in the HER, which were assessed by employing graphite counter electrode. There are not so many alternatives to the platinum catalyst in HER, yet the system described in this work may stimulate researchers to seek for cheaper, yet, high performing alternatives. It's just a pity that the approach proposed by the authors could not be explored on the industrial scale.

Overall, this manuscript is well written and illustrated. Multiple complementary techniques have been employed to gain a thorough understanding on these hybrid van der Waals heterostructures and their properties. The data analysis has been carried out with care and the conclusions are well supported by the experimental and computational evidences.

Overall, this paper reports a major step forward on the controlled modulation of the electronic properties of 2D semiconductors when interfaced with supramolecular systems. I believe it will appeal the broad audience of Nature Communications, therefore I recommend its acceptance pending minor revisions.

1) The STM images of the mono-component networks is missing.

2) In the HER experiments: how important is the order of the layers ? What happens if you change the order of layers (2DMs vs SAM).

Point by point reply

Reviewer #1

General comments R1: *This manuscript reported that the monolayer hydrogen-bonded 2D network of cyanuric acid (CA) and melamine (M) can be adsorbed on MoS₂ and WSe₂ to form a hybrid organic/inorganic van der Waals heterostructure. By virtue of the ultra-strong n-doping effect of CA-M, MoS₂ showed an exponential increase in charge carrier density and a higher field-effect electron mobility. Moreover, the hybrid heterostructure can be applied to acidic HER, which greatly improved the activity of MoS₂ (even better than the benchmark Pt).*

Reply to the general comments R1: We are grateful to Reviewer #1 for his/her positive assessment of our work, and for recognizing the novelty of our work.

Comment R1_1: *Although the application of CA-M hydrogen-bonded layers to enhancing activity of HER catalysts have been reported (Adv. Mater. 2021, 33, 2007368) but not discussed or cited, this work mainly focused on the fields of semiconductors and FET, which has enough novelty for publication in Nat. Commun.*

Response to the Comment R1_1: We fully agree that the manuscript Adv. Mater. 2021, 33, 2007368 should be included in the list of references and discussed in the paper, yet it does not compromise the novelty of our work.

Our action on general comments R1: We have now included the article Adv. Mater. 2021, 33, 2007368 as reference no. 54 and discussed it in the Main Text at page 10.

Comment R1_2: *The thickness of CA-M SAMs was claimed as 0.48 nm. However, according to crystallography, the thickness of monolayer CA-M should be about 0.31 nm. Revisions and/or reasonable explanations are needed.*

Reply to R1_2: We thank the Reviewer for this comment which triggered us to perform a statistical analysis on a larger pool of data. To quantify the thickness of the CA-M monolayer, we performed AFM imaging of sub-monolayer thick films (Figure P1a); the CA-M layer thickness determined from 20 different topographical profiles amounts to 0.397 ± 0.103 nm. It is important to note that the estimation of the height of a single layer of two-dimensional material's flake and/or single-atom-thick self-assembled monolayer from profiles extracted from topographical AFM images depends on the employed substrate and the environmental conditions such as relative humidity. Moreover, our measurements revealed that the bare MoS₂ surface has a root mean square roughness of about 0.1 nm as determined on an area of $10 \mu\text{m} \times 10 \mu\text{m}$ (see Figure P1b), which may also introduce measurement errors.

Figure P1: (a) Statistics of the thickness of CA·M molecular layer on sub-monolayer samples. (b) the roughness of the MoS₂ crystal surface. (c) side view of the simulated structure of CA·M monolayer adsorbed on MoS₂ surface.

Noteworthy, the distance between the two planar sheets in a CA·M crystal has been previously estimated by AFM imaging as 0.317 nm (CrystEngComm, 2011, 13, 1067–1069). However, great care should be taken when comparing the latter experiment with the one reported in our paper because of the differences in the local environment of the 2D CA·M network. More precisely, while in our work the CA·M monolayer was physisorbed onto the MoS₂ surface hence it encountered strong molecule-substrate interactions, in the CrystEngComm 2011 paper the investigated CA·M layers are stacked via van der Waals forces forming crystals. The different substrates can also lead to various adsorption distances, as our simulated vertical distance between CA·M molecular layer and MoS₂ surface is about 0.33 nm (see Figure P1c), while the simulated average distance between CA·M molecular layer and Au (111) surface has been reported being 0.371 nm (J. Phys. Chem. C 2008, 112, 4209-4218).

Our action on R1_2: In the revised version of the Supplementary Information at page S2 we have added the AFM results as Fig. S3c-d. The discussion of CA · M monolayer thickness dependent on local environments has been added in Supplementary Information on page S3: “*In order to quantify the thickness of the CA·M monolayer more accurately, we performed AFM topographical imaging of sub-monolayer thick films. Fig. S3c displays the statistics on the CA·M layer thickness executed by sampling data from 20 different sample’s regions; the resulting average value amounts to 0.397 ± 0.103 nm. Moreover, the bare MoS₂ surface has a roughness of about 0.1 nm, as depicted in Fig. S3d. Noteworthy, the distance between the two planar sheets in CA·M crystal has been previously estimated by AFM imaging as 0.317 nm. The difference in the CA·M monolayer thickness is mostly caused by the local environment. More precisely, while in our work the CA·M monolayer was physisorbed on MoS₂ surface hence it encountered strong molecule-substrate interactions, in the melamine–cyanuric acid crystal, the investigated CA·M layers are stacked via van der Waals forces. The use of different substrates can also lead to various adsorption distances, as our simulated*

vertical distance between CA·M molecular layer and MoS₂ surface is about 0.33 nm, while the simulated average distance between CA·M molecular layer and Au (111) surface has been reported being 0.371 nm.”

Comment R1_3: *It is better to directly characterize the valence state of the materials to support the claim of n-doping. Besides, a deeper explanation for enhancement of HER activity after n-doping should be given.*

Reply to R1_3: We thank the Reviewer for her/his comment. The catalytic activity of MoS₂ can be affected by the presence of buffer layers in two ways: *i*) via the formation of an interfacial tunneling barrier with MoS₂ or *ii*) through the modification of the chemical nature of MoS₂ *via* charge transfer (i.e., proximity doping). For the former, a metallic substrate generating a low interfacial tunneling barrier with MoS₂, such as Ti, is used (ACS Nano 2020, 14, 2, 1707–1714). For the latter case, molecular *n*-type doping processes effectively increase the charge carrier density in MoS₂, which can improve the catalytic ability of active sites (S vacancies) on the MoS₂ surface, and our CA·M film is suitable for this case. The catalytic active sites (S vacancies) exist on the MoS₂ surface regardless of the presence of an underlying CA·M buffer layer. Thin CA·M molecular film has no obvious hydrogen evolution catalytic performance, and it only acts as a strong *n*-dopant for the top MoS₂ layer. In HER, the adsorbed protons sequester electrons from the MoS₂ surface thereby forming hydrogen molecules through a process of desorption from the surface. Therefore, the electron (charge) density on the MoS₂ surface can affect the rate of hydrogen evolution reaction to a certain extent. The electron transfer in the CA·M / MoS₂ heterostructure occurs from CA·M to MoS₂, hence increasing the charge density on the MoS₂ surface, and further enhancing the HER activity.

To gain a better understanding of the *n*-doping mechanism of MoS₂ by CA·M layer adsorption, ultraviolet photoelectron spectroscopy (UPS) and X-ray photoelectron spectroscopy (XPS) measurements have been performed in order to cast light onto the interfacial electronics of the monolayer MoS₂ / CA·M heterostructure interface. A comparison of the recorded UPS spectra before and after CA·M film (about 2 nm thick) deposition is shown in Figure P2. The linear extrapolation of the low kinetic energy onset reveals that the work function of monolayer CVD MoS₂ sharply decreases from 4.0 eV to 1.6 eV (Figure P2b), as a result of the substantial interfacial electron transfer from CA·M layer to MoS₂. At the low energy region, the work function reduction is accompanied by a downward band-bending of the valence band of MoS₂ towards higher binding energy. In the low binding energy region of UPS spectra (Figure P2c,d), the binding energy at valence band maximum upshifts by 0.33 eV (from 1.85 eV to 2.18 eV) away from Fermi level (E_F). Figure P3a,b shows the evolution of the Mo 3d and S 2p core levels with the increased CA·M coverage. At the initial stage, with sub-monolayer coverage (drop-casting 1 × 10⁻⁸ M solution), the binding energies of Mo 3d and S 2p core levels slightly increased by ca. 0.15 and 0.13 eV, respectively. This indicates that the Fermi level of MoS₂ shifts toward the conduction band, hence confirming the effective electron doping process. With an increase in the CA·M coverage to a complete monolayer (drop-casting 1 × 10⁻⁷ M solution), the binding energies of Mo 3d and S 2p core levels significantly decrease by about 0.26 and 0.27 eV, respectively. A further increase in the CA·M thickness to around two layers (2nd drop-casting 1 × 10⁻⁷ M solution) yields a decrease of

binding energies of 0.40 eV for Mo 3d and 0.41 eV for S 2p. The decrease of the core level energy is caused by the extra electron-induced phase transition of TMDs (ACS Nano 2018, 12, 2070-2077). The transfer curve (Figure P3c) demonstrates that the CA·M coated monolayer MoS₂ exhibits n-type doping characteristics. Moreover, when the channel is fully covered with CA·M film (drop-casting 1 x 10⁻⁷ M solution), very weak gate dependence is observed, revealing a transition from the semiconducting phase to the metallic phase. Herein, we have calculated the electron conductance ($G = I_{sd}/V_{sd}$) at $V_g = 0$ V with the increased CA·M thickness, as shown in Figure P3d. The electron conductance of monolayer MoS₂ increase from 0.6 to 130 μ S, which confirms the strong electron doping on MoS₂.

Figure P2: (a) UPS spectra of monolayer MoS₂ before and after CA·M film attachment. (b) low kinetic energy (secondary electron cutoff) region. (c-d) low binding energy region (near the EF) of MoS₂ (c) and MoS₂ / CA·M (d).

Figure P3: XPS core-level spectra of (b) Mo 3d and (c) S 2p as a function of CA·M thickness on monolayer MoS₂. (c) Transfer characteristics of monolayer MoS₂ device with increasing CA·M film from sub-monolayer to around bilayer, V_{ds} = 0.1 V. (d) Electric conductance at V_g = 0 V based on the device shown in (c).

Our action on R1_3: In the revised version of the Main Text on page 10 we have added “In HER, the adsorbed protons sequester electrons from the MoS₂ surface thereby forming hydrogen molecules through a process of desorption from the surface. Therefore, the electron density on the MoS₂ surface can indirectly affect the rate of hydrogen evolution to a certain extent. The electron transfer in the CA·M / MoS₂ heterostructure occurs from CA·M to MoS₂, hence increasing the charge density on MoS₂ surface and further enhancing the HER activity.”

In the revised version of the Supplementary Information on page S19, we have added the UPS and XPS results as Fig. S24.

Comment R1_4: *It is better to clearly explain the advantages and disadvantages of applying a gate voltage for HER, particularly in the viewpoint of practical application.*

Reply to R1_4: We fully concur with Reviewer #1 that it is worth explaining more carefully the advantages and disadvantages of applying a gate voltage for HER.

Advantages: the additional electric field can increase the channel conductance of the MoS₂ nanosheet-based HER devices, which can effectively enhance MoS₂'s catalytic performance with faster charge transfer. Such a strategy of utilizing the electric field to improve the catalytic activity of HER could also be employed in the other electrochemical processes.

It is important to note that the back gating provides a degree of freedom that is not generally available in conventional electrocatalysis experiments. In a conventional electrocatalysis experiment, double-layer charging of active sites surely occurs but there is no ability to reversibly

change the degree of charging independently of the electrochemical potential. When applying a back gate on sufficiently thin electrocatalyst layers, the number of charges in the active sites is independently controlled and the impact on the overpotential can then be assessed directly and immediately.

Disadvantages: four electrodes system is more complicated than three electrodes system in cyclic voltammetry experiments, hindering to some extent our gate-enhanced HER strategy for large-scale technological applications.

Our action on R1_4: In the revised version of the Main Text on page 12 we have added “Such a strategy of utilizing the electric field to improve the catalytic activity of HER could also be used in the other electrochemical processes. However, the use of more complicated four electrodes system in cyclic voltammetry experiments may hinder to some extent this gate-enhanced HER strategy for large-scale technological applications.”

Reviewer #2

General comments R2: *This paper reports an apparently important result for electrocatalysis, namely that formation of a MoS₂/organic van der Waals heterostructure (VDWH) allows a dramatic reduction in the overpotential for the hydrogen evolution reaction (HER) on the MoS₂. However, I have numerous criticisms of this paper and I do not think it is of suitable quality for Nature Communications. My specific comments are below.*

Reply to the general comments R2: We thank Reviewer #2 for taking the time to assess our manuscript. We have now addressed extensively all the points that were raised.

Comment R2_1: *The voltammetry shown in Figure 5c is indeed impressive. However, the authors offer no mechanism for enhancement of electrocatalytic activity...they simply ascribe it to excess electron density. That idea alone is not a mechanism or an explanation. The idea that putting electrons into MoS₂ using a gate stack has been published before.*

Reply to R2_1: We thank the Reviewer for this comment and for giving us the chance to provide a deeper interpretation of the process ruling the observed enhancement of the electrocatalytic activity. We would like to point out that when compared to the bare MoS₂, the CA·M / MoS₂ heterostructure displays an improvement in HER performance also in absence of an applied gate voltage (Fig. S20c). The unique property of our novel heterostructure relies on the strong *n*-doping effect imparted by the CA·M superlattice on the monolayer MoS₂. By virtue of the positive cooperative effect in bi-component CA·M SAM, a charge transfer magnitude as high as 10¹³ level could be obtained, which is notably high when benchmarked with the analog effect induced by strong organic dopants. Such a strong *n*-doping effect is evidenced by detailed FET studies in which the CA·M supramolecular layer is placed on the top (Fig. 3b) or the bottom (Fig. S21) of MoS₂. The HER application of the CA·M/MoS₂ heterostructure represents a proof-of-concept that the strong *n*-doping effect can effectively increase the electron density on the MoS₂ surface. Moreover, in our device one can take

advantage of a positive gate voltage that, when applied, it can increase channel conductance and enhance the proton adsorption on active sites (S vacancies). By combining the strong *n*-doping effect from CA·M layer and applied positive back gate voltage, a stark enhancement in the catalytic activity of monolayer MoS₂ for HER is observed, outperforming the platinum catalyst.

Comment R2_2: *Indeed, the authors note reference 54 which is the first report of the field-effect enhancing HER at MoS₂ sheets. However, the conclusion of reference 54 is that the reason for the enhancement of activity is simply that the contact resistance between the metals and the MoS₂ is reduced on gating...this is a rather mundane result. Likewise, reference 56 makes the same conclusion...this time that electrolyte gating of a MoS₂ flake lowers the contact resistance between the metals and the MoS₂. Neither paper argues for an INTRINSIC increase in the activity of the active sites on MoS₂ (S vacancies). The authors should instead look at Y. Wang 2019, 19, 6118...that paper proposes an explicit mechanism for HER activity increase with gating (along with experimental results). That paper is directly relevant to the authors' current work. Their results should be compared to it.*

Reply to R2_2: We thank Reviewer #2 for bringing to our attention the manuscript Nano Lett. 2019, 19, 6118-6123) which we had overlooked in the original version of our manuscript. It is now cited (see reference 59, page 18 in the Main Text).

S defects undeniably play a key role in MoS₂-based optoelectronic devices. Because of its defective nature, MoS₂ behaves as an *n*-type semiconductor. On the same time, the HER catalytic activity of MoS₂ is also enhanced by the presence of S defects. In addition to the mechanism of increasing channel conductivity, Wang *et al.* proposed that gate-induced charging of the localized d-state of unsaturated Mo in S vacancies (the active sites), which is very helpful to explain our HER results. In our work, we mostly focus on the influence of the CA·M buffer layer on the MoS₂ HER catalytic activity. The interfacing MoS₂ with CA·M does not alter the number of catalytic active sites (S vacancies) per unit area on MoS₂, being independent of the presence of a CA·M buffer layer placed on top or below the MoS₂ sheet. To simplify the theoretical model, the defects on MoS₂ have not been considered when simulating the charge transfer process in CA·M / MoS₂ heterostructure. We observed a notable improvement in the HER performance in the CA·M / MoS₂ heterostructure when compared to the neat MoS₂ sheet, which according to our findings originates from the strong *n*-doping effect from CA·M layer. In the presence of the CA·M buffer layer the overpotential decreased by 130 mV when no gate voltage is applied. Interestingly, such an improvement in the heterostructure is even more pronounced when a positive gate voltage was applied (Fig. S23).

The HER performance of our monolayer MoS₂ devices (supported on SiO₂ substrate) is generally consistent with the results reported in Wang's paper. The overpotential and Tafel slope of MoS₂ sheets both decrease with increasing back gate voltage. The difference in exact value comes from the different experimental parameters. In Wang's work, 300 nm thick SiO₂ on Si was used as substrate and CVD MoS₂ (lateral size > 100 μm²) flakes as working electrodes. Conversely, in our work, the SiO₂ layer is 270 nm thick, which may cause enhancement of gate modulation.

The capacitance of 270 nm thick SiO₂ (13 nF/cm²) is greater than 300 nm SiO₂ (10 nF/cm²), and a larger dielectric layer capacitance can cause a smaller subthreshold swing in FET devices. A FET device characterized by a steep subthreshold slope exhibits a faster transition between off (low

current) and on (high current) states, which means that upon applying the same V_{GS} and V_{DS} , a smaller subthreshold swing leads to a larger I_{DS} increasing. Compared to 300 nm thick SiO_2 substrate, the same gate voltage can induce a larger current density of MoS_2 sheets supported on a 270 nm thick SiO_2 substrate. The monolayer MoS_2 flakes we used for HER are mechanically exfoliated from the crystal (lateral size of 1-10 μm^2), which have different surface defect ratios and circumference/area ratios compared with the CVD flake.

We also attempted to modulate the HER performance of thick MoS_2 sheets by applying different voltages across the gate and source terminals (V_{GS}), as displayed in Figure P4 below. Consistent with the assumption reported in Wang's paper, the enhancement is not significant.

Figure P4: (a) the optical image of 35 nm thick MoS_2 device. (b) The polarization curves for HER on MoS_2 flake under different gate voltages.

Our action on R2_2: We have now included the article Nano Lett. 2019, 19, 6118-6123 as reference no. 59 and discussed it in the Main Text on page 12: “Moreover, when MoS_2 surface is immersed in a sulfuric acid electrolyte, the applied positive gate voltage has also been confirmed to induce the excess electron density on the Mo atoms near the S vacancy sites, leading to enhanced Mo-H bond strengths (stabilizing H proton adsorption)”

Comment R2_3: *The presentation of the paper is very frustrating. The introduction does not read well...and even though the HER result is really the most significant finding, there is no hint of this result in the title of the paper.*

Reply to R2_3: We thank Reviewer #2 for this suggestion, and we do agree that the HER results should be highlighted in the title of our manuscript. Since the take-home message of our work is essentially the controlled formation of supramolecular 2D networks held together by non-covalent interactions and their physisorption onto 2D semiconductors forming hybrid heterostructures, in the introduction of our paper we paid a special attention to these concepts and discussed their state-of-the-art. We opted not to include a long paragraph on HER in 2D materials because the HER represents only one among other possible applications of our $\text{MoS}_2/\text{CA} \cdot \text{M}$ heterostructure. We appreciate the positive evaluation from the Reviewer about the catalytic performance of our $\text{MoS}_2/\text{CA} \cdot \text{M}$ heterostructure. The previous works on MoS_2 based van der Waals heterostructures for HER are also highlighted in the introduction section.

Our action on R2_3: In the revised version of the Main Text on page 3 we have added “The studies on MoS₂ as an earth-abundant and inexpensive material are essential for the implementation of clean energy technologies using hydrogen. The formation of MoS₂ based 2D heterostructures, e.g., graphene / MoS₂²⁶ and WS₂ / MoS₂²⁷, induces a built-in electric field formed among the dissimilar layers, enhancing the MoS₂ catalytic performance.”

Furthermore, the title of our manuscript has been changed into: “Dressing 2D semiconductors with supramolecular 2D hydrogen-bonded superlattices: boosting the electronic and catalytical properties of MoS₂ via cooperative effects”

Comment R2_4: *What is the mechanism for the charge transfer between the organic layer and the MoS₂? At the top of page 10 the authors state that "The n-type doping on MoS₂ mostly originates from the charge transfer effect of the CAM molecular network" but what does that mean? What molecule gets oxidized M or CA? Would M alone dope the MoS₂? Why not?*

Reply to R2_4: We thank Reviewer #2 for this interesting question and for giving us the opportunity to clarify further the mechanism of charge transfer.

The photo-induced charge transfer process involves the gains and losses of electrons, which can be regarded as a redox process (Science, 1992, 258, 1474-1476). The electrical conductivity (σ) of semiconductors is calculated as $\sigma = n * q * \mu$. As the mobility (μ) doesn't change much, the doping-induced increase of σ is mainly caused by the substantial increase of carrier concentration (n). TMDs' electronic properties - the electron and hole concentration - can be modulated *via* non-covalent modification with organic molecules, and this phenomenon, when using molecules possessing a modest intrinsic dipole moment, is generally understood as charge transfer occurring at the semiconductor/molecule interface (Chem. Soc. Rev., 2018, 47, 6845. ACS Nano 2019, 13, 9713-9734).

In our work, the charge transfer process from the organic layer to MoS₂ has been elucidated by means of theoretical calculations. Figure P5a displays the charge density difference plot, (upper part: top view, bottom part: side view). The charge density difference plot has been calculated as the difference between the total charge density of CA·M unit cell adsorbed on MoS₂ and the combined charge density of isolated CA·M unit cell and MoS₂ lattice. Charge depletion (blue) and accumulation (yellow) take place distinctly between the interface of monolayer MoS₂ and the CA·M superlattice. Compared with the clean MoS₂, the interfacing with CA·M causes an electron enrichment on the MoS₂ surface. Furthermore, the theoretical value of the charge transfer intensity can be quantified directly as $3.62 \times 10^{13} \text{ e/cm}^2$.

Figure P5b displays the lone pair electrons of nitrogen on both M and CA molecules, and these lone pair electrons can be delocalized to dope MoS₂. For the M molecule, the hybridization of N in the nitrogen-containing heterocycle is SP², and the hybridization of N in amino is SP³. The SP³ hybridized N has a stronger ability to donate electrons than SP² hybridized N, so the lone pair electrons in the amino groups on M are most likely to be delocalized. The basal S atoms above the amine groups are more active in MoS₂ HER has been reported (J. Mater. Chem. A, 2019, 7, 22571–22578). CA molecule tautomerizes between triacid and triketone, and the hybridization of N is switchable between SP² and SP³. Therefore, the lone pair electrons on the CA molecule can also be delocalized and form an *n*-doping effect on MoS₂.

In Fig.1 of the Main Text, the calculation results have demonstrated that both M and CA molecules can transfer electrons to MoS₂. Because of the weaker intermolecular hydrogen bonds (compared with CA·M co-assembly), neither M nor CA can form ordered 2D superlattices on the MoS₂ surface, leading to a weaker doping effect performance in FET devices, because of the absence of a cooperative effect (Fig. S13).

Figure P5: (a) the charge density difference plot of MoS₂ / CA·M heterostructure, upper part: top view, bottom part: side view. (b) lone pair electrons of nitrogen on M (upper) and CA (bottom) molecules.

To assess the involvement of MoS₂ / CA·M, MoS₂ / M, and MoS₂ / CA interfaces in the charge transfer processes, core level spectra (X-ray photoelectron spectroscopy, XPS) of monolayer CVD MoS₂ with sub-monolayer molecular coverage are analyzed (Figure P6). Note, the N 1s level is related to CA and M molecules. Compared with molecular film on SiO₂, the N 1s core-level shift to higher energy for all three interfaces, which dominantly represents the process of losing electrons from CA and M molecules owing to the interfacial charge transfer (Adv. Mater. 2021, 33, 2008752). The MoS₂ / CA·M interface has the most obvious shift of N 1s by 0.7 eV, compared with 0.4 eV for MoS₂ / M interface and 0.35 eV for MoS₂ / CA interface. In the case of CA, the N 1s XPS spectra could be deconvoluted into two peaks, as seen in Figure P6c. The peak around 400.5 eV is assigned to the nitrogen of CA tautomer in the form of triketone, whereas the peak around 398.7 eV relates to the nitrogen of CA tautomer in form of triacid. The charge transfer is more obvious when the CA molecule is adsorbed on the MoS₂ surface in triacid form. Therefore, for all three systems, the N 1s core-level spectra consistently confirm the occurrence of electron transfer from molecule layer to MoS₂.

Figure P6: N 1s XPS core level spectra for (a) MoS₂ / CA·M interface, (b) MoS₂ / M interface and (c) MoS₂ / CA interface.

Please also read our reply to **R1_3** in which we explained the effect of charge transfer to MoS₂ surface revealed by an in-depth UPS and XPS analysis.

Our action on R2_4: In the revised version of the Supplementary Information on page S20 we have added the XPS results as Fig. S25.

Comment R2_5: *CPD vs SP. In Kelvin probe, the difference in work function between the tip and the sample is called the contact potential difference, not the surface potential. The authors use CPD in Fig. 4 and then the text speaks of surface potential. This should be changed.*

Reply to R2_5: We agree with Reviewer #2 that for the sake of consistency, the contact potential difference (CPD) is the correct term to be used.

Our action on R2_5: The sentence “The SP scale was set to zero at the $E_{F,Au}$ value to facilitate the visual comparison of the SP of MoS₂ before and after CA·M doping.” has been deleted from the captions of Figure 4.

In the revised version of the Main Text on page 9, we have included a new sentence: “The evolution of the WF of a monolayer MoS₂ surface is measured by Kelvin probe force microscope (KPFM), which allows recording contact potential map.”

Comment R2_6: *The point on cooperative effects is not clear enough. Why does formation of the 2D molecular assembly allow doping of the MoS₂ substrate?*

Reply to R2_6: As clearly defined by Hunter and Anderson in their Essay (Angew. Chem. Int. Ed. 2009, 48, 7488 – 7499), “cooperativity arises from the interplay of two or more interactions so that the system as a whole behaves differently from expectations based on the properties of the individual interactions acting in isolation.” Hence, cooperativity is a key feature in systems chemistry that leads to collective properties otherwise not present in the individual molecular components. Positive cooperativity displays unique properties which are enhanced as a result of the ensemble, meaning that the effect is outperforming the one resulting from the sum of the isolated individuals. In this work, the charge transfer from the molecular adsorbate to the MoS₂ displays a magnitude that varies

linearly with the degrees of coverage (i.e., number of molecules adsorbed on the MoS₂ surface per unit area) for monocomponent networks of M or CA whereas it displays a non-linear (exponential) trend when the adlayer comprises the bi-component CA-M network. This is a positive cooperative effect that induces a strong charge transfer that ultimately leads to a major modification of the electronic properties of the 2D semiconductors. This positive cooperativity was evidenced computationally (Fig. 1b, Main Text).

Reviewer 3

General comments R3: *The manuscript by Wang and co-workers reports the formation of inorganic/organic van der Waals heterostructures based on MoS₂ functionalized with cyanuric acid-melamine (CA-M) based extended 2D supramolecular superlattices. It shows that the interfacing of the supramolecular self-assembled network of CA-M with MoS₂ determines a strong n-doping via charge transfer as evidenced by a markedly high increase in charge carrier density, exceeding 10¹³ cm⁻². Such a charge transfer from the physisorbed CA-M 2D networks to the MoS₂ and/or WSe₂ determines an upward shift of the Fermi level of the two-dimensional material. Not surprisingly, the CA-M layer has been also found to determine a significant improvement in the electronic transport through the WSe₂. The experimental data has been corroborated by DFT calculations.*

Although, the idea of forming 2D self-assembled monolayers based on cyanuric acid and melamine is not new (following the pioneering works of Whitesides in 1990, as detailed by the authors), its use – via controlled interfacing – for modulating the electronic properties of 2D semiconductors is unprecedented. The reported findings open many perspectives on the use of a variety of chemically tailored and atomically precise 2D bi- and multi-component supramolecular superlattices which have been reported during the last 4 decades for tuning the electronic properties of 2D semiconductors via controlled doping.

From a more fundamental viewpoint, is extremely interesting that the while monocomponent networks of melamine and cyanuric acid mildly affect the electronic properties of the 2D semiconductors, the bi-component CA-M network causes major changes in the charge transfer in a process featuring a cooperative nature. This unique phenomenon was clearly unraveled computationally.

I was also impressed by the performance of the CA-M modified 2D materials in the HER, which were assessed by employing graphite counter electrode. There are not so many alternatives to the platinum catalyst in HER, yet the system described in this work may stimulate researchers to seek for cheaper, yet, high performing alternatives. It's just a pity that the approach proposed by the authors could not be explored on the industrial scale.

Overall, this manuscript is well written and illustrated. Multiple complementary techniques have been employed to gain a thorough understanding on this hybrid van der Waals heterostructures and their properties. The data analysis has been carried out with care and the conclusions are well supported by the experimental and computational evidences.

Overall, this paper reports a major step forward on the controlled modulation of the electronic properties of 2D semiconductors when interfaced with supramolecular systems. I believe it will appeal the broad audience of Nature Communications; therefore, I recommend its acceptance pending minor revisions.

Reply to the general comments R3: We thank Reviewer #3 for the very positive assessment of our work.

Comment R3_1: *The STM images of the mono-component networks is missing.*

Reply to R3_1: Since the mono-component, CA or M, cannot form ordered self-assembled monolayers on the MoS₂ surface, it is not possible to visualize them with STM under ambient conditions. To compensate for this, we have monitored by AFM imaging both mono-component films (Fig. S11 and S12). We found that both CA and M molecules form aggregates on the MoS₂ surface, with CA molecules being more strongly aggregated.

Comment R3_2: *In the HER experiments: how important is the order of the layers? What happens if you change the order of layers (2DMs vs SAM).*

Reply to R3_2: This is an interesting question. In the CA·M/MoS₂ heterostructure, CA·M molecular film has no obvious HER catalytic, and it only acts as a strong n dopant for the top MoS₂ layer, whereas the HER occurs on the MoS₂ surface. Hence, the MoS₂ surface needs to be exposed to the electrolyte as a working electrode for the hydrogen evolution reaction. If the CA·M layer is on top of MoS₂, some active sites on the MoS₂ surface will be covered, leading to suppression in its HER efficiency, as seen in Figure P7.

Figure P7: The HER polarization curves of monolayer MoS₂, monolayer MoS₂ on CA·M thin film, and CA·M thin film on monolayer MoS₂.

REVIEWERS' COMMENTS

Reviewer #1 (Remarks to the Author):

The authors addressed my comments properly.

Reviewer #2 (Remarks to the Author):

I think the authors have largely satisfied the comments of the reviewers. I still have questions about the gate field effect on top of the doping and how such a small gate voltage (5V) can effect a substantial change in the I-V characteristics when the gate capacitance is only 13 nF/cm². But the authors have improved the paper and I am OK with publication at this stage.

Reviewer #3 (Remarks to the Author):

The authors have addressed all my concerns. I strongly recommend its publication in Nature Communications.

Reviewer #2

Comment R2: *I think the authors have largely satisfied the comments of the reviewers. I still have questions about the gate field effect on top of the doping and how such a small gate voltage (5V) can effect a substantial change in the I-V characteristics when the gate capacitance is only 13 nF/cm². But the authors have improved the paper and I am OK with publication at this stage.*

Reply to R2: We thank Reviewer #2 for taking time to re-assess our manuscript, giving an overall positive assessment of our work.

From the value mentioned (5V), the Reviewer refers to the gate potential applied during the HER experiments. Thus, the I-V characteristics that are referred to undergo substantial changes are those related to electrochemical signal, i.e., the current flow across the MoS₂/electrolyte interface (from the working electrode to the counter electrode), and not the transistor channel current (I_{DS}), from source and drain electrodes. When referring to the gate capacitance (C_i), I_{DS} is the one that is directly proportional to C_i (see Eq. 1 in the manuscript), and indeed a gate potential increase of Δ5V does not provide substantial I_{DS} changes. That is because I_{DS} also depends on the carrier mobility, contact resistance, possible Fermi level pinning and/or Schottky barrier at the contacts that may jeopardize the transistor performance. However, when applying 5V as gate voltage the transistor is clearly in ON state (Supplementary Figure 13d), meaning that carriers are available within the MoS₂ flake and any charge carrier recombination caused by the oxide interface is overcome while applying such a gate potential. The excess of charges in the MoS₂ flake induced by the 5V gate potential can be calculated by $\Delta n_{BG} = C_i \times V_{BG} / q = 4 \times 10^{10} \text{ cm}^{-2}$, which is related to the capacitance of SiO₂ layer. Indeed, the supplementary data below shows the MoS₂ transistor transfer curve without the CA·M layer (a) and with the CA·M layer (b). It reveals that the channel current change ΔI_{DS} is higher when the CA·M supramolecular layer is present. If we calculate the excess of charges in the respective sample from the transfer curves using $\Delta I_{DS} = \mu \times q \times \Delta n$, we find Δn is $5.1 \times 10^{10} \text{ cm}^{-2}$ for MoS₂ device (Fig. P1a), and $1.0 \times 10^{11} \text{ cm}^{-2}$ for CA·M / MoS₂ heterostructure device (Fig. P1b). Δn is not the same value with Δn_{BG} when the CA·M is employed, meaning that the supramolecular layer plays an important role while increasing the electron density in the functionalized MoS₂.

Figure P1. Transfer characteristics of MoS₂ FET (a) and CA·M / MoS₂ heterostructure FET (b) on Si / SiO₂ substrate. I_{DS} is the drain current and V_{GS} is the gate potential.

A 5V gate voltage across the SiO₂ dielectric establishes an electric field (E) of 18.5 MV/m ($V = E \cdot d$) that is high enough to increase the density of carriers in the MoS₂ available for HER at the interface with the electrolyte, thereby increasing the electrochemical signal, which is further boosted when the CA·M layer is present. Notice that the MoS₂/electrolyte double-layer capacitance (tens of $\mu\text{F}/\text{cm}^2$) is the one that governs the HER reaction, while the oxide dielectric capacitance serves to control the transistor operation. In HER measurements, the hydrogen evolution is markedly influenced by the efficiency of electron injection from the substrate to the catalyst and its transport to the active site. For monolayer MoS₂, the possible strong electrical coupling between it and the substrate may lower the contact resistance to facilitate charge injection from the substrate to the catalysts, further boosting the gate-enhanced catalytic performance. The further mechanism will be studied in a follow-up work. The gate-assisted modulation of the HER has becoming a common practice in 2D material-based devices (Adv. Mater. 2017, 29, 1604464; Nano Lett. 2019, 19, 8118-8124), whose values of gate voltage typically applied are in accordance with those used in this work.